# Association of lipoprotein (a) with coronary artery disease in a South Asian population: A case-control study

**Jacob Joseph[1], Jaideep Chanayil Menon****[2]\*, Placid K. Sebastien[3], Abish Sudhakar[4], Denny John[5], Geetha R. Menon****[6]\***

**1** Department of Cardiology, Lisie Hospital, Ernakulam, Kerala, **2** Amrita Institute of Medical Sciences & Research Centre, Kochi, Kerala, **3** Department of Cardiology, Aster MIMS, Chala, Kerala, **4** Department of Pediatric Cardiology, Amrita Institute of Medical Sciences & Research Centre, Kochi, Kerala, **5** Adjunct Faculty, Public Health, Amrita Institute of Medical Sciences & Research Centre, Kochi, Kerala, **6** ICMR-National Institute of Medical Statistics, New Delhi, India

\* menon7jc@gmail.com (JCM); menongr.hq@icmr.gov.in (GRM)

## Abstract

### Introduction

Coronary artery disease (CAD), the leading cause of mortality worldwide, is characterised by an earlier onset and more severe disease in South Asians as compared to Western populations.

### Methods

This is an observational study on 928 individuals who attended three tertiary care centres in Kerala, India from 2014-to 2017. The demographic, anthropometric, behavioural factors and the lipoprotein (Lp(a)) and cholesterol values were compared between the two groups and across disease severity. The Chi-square test was used to compare the categorical variables and independent sample t-test for the continuous variables. Multivariable logistic regression was performed to investigate the association of demographic, clinical and behavioural factors with the disease. Odds ratios are presented with a 95% confidence interval. In individuals below 50 years, two logistic regression models were compared to investigate the improvement in modelling the association of the independent factors and Lp(a) with the occurrence of the disease.

### Results

We included 682 patients in the diseased group and 246 patients treated for non-coronary conditions in the control group. Those in the control group were significantly younger than in the diseased group(p<0.002). Significantly more patients were diabetic, hypertensive, tobacco users and consumers of alcohol in the diseased group. Multivariable logistic regression on data from all age groups showed that age (OR = 2.55, 95% CI 1.51–4.33, p = 0.01), diabetes (OR = 3.71, 95% CI 2.42–5.70, p = 0.01), hypertension (OR = 3.03, 95% CI 2.12–4.34, p = 0.01) and tobacco use (OR = 5.44, 95% CI 3.39–8.75, p = 0.01) are significantly

**Data Availability Statement:** The supporting data is provided in the Supporting information file.

**Funding:** The authors received no specific funding for this work.

**Competing interests:** The authors have declared that no competing interests exist.

associated with the disease. Lp(a) (OR = 1.22, 95% CI 0.87–1.72) increased the odds of the disease by 22% but was not statistically significant. In individuals below 50 years, Lp(a) significantly increased the likelihood of CAD (OR = 3.52, 95% CI 1.63–7.57, p = 0.01). Those with diabetes were seven times more likely to be diseased (OR = 7.06, 95% CI 2.59–19.21, p = 0.01) and the tobacco users had six times the likelihood of disease occurrence (OR = 6.38, 95% CI 2.62–15.54, p = 0.01). The median Lp(a) values showed a statistically significant increasing trend with the extent/severity of the disease in those below 50 years.

## Conclusion

Age, diabetes, hypertension and tobacco use appear to be associated more with the occurrence of coronary artery disease in adults of all ages. Lipoprotein(a), cholesterol and BMI categories do not seem to be related to disease status in all ages. However, in individuals below 50 years, diabetes, tobacco use and lipoprotein (a) are significantly associated with the occurrence of the disease.

## Background

Hyperlipidaemia is one of the major risk factors for ASCVD (atherosclerotic CVD), with elevated levels of low-density lipoprotein cholesterol (LDL-C) being a major contributor to subsequent CVD events [1, 2]. Despite numerous clinical trials have shown that reducing LDL-C levels, substantially reduces the risk of CVD suggesting a strong causality, individuals continue to have residual CVD risk and suffer from CVD events despite significant LDL-C lowering in addition to the fact that many individuals have ASCVD despite normal lipid values [3]. There are very likely other factors influencing atherosclerosis of which lipoprotein (a)- [Lp(a)], is a likely candidate, especially in the young coronary artery disease (CAD) subset of patients defined as males below 55 years and females below 65 years of age with CAD [4–6].

Previous studies have published a robust association between Lp(a) and CVD outcomes in the general population. A wealth of current evidence suggests that an increased Lp(a) level is associated with a modest increase in the risk of future CVD events in both general and high-risk populations. Such an association of Lp(a) with CVD, is independent of LDL, reduced high-density lipoproteins (HDL), and other traditional CVD risk factors [7, 8].

Lp(a) is uninfluenced by age, sex, diet, or environmental factors, with stable lifelong levels being attained by age of two [9–11]. Lp(a) levels have shown worldwide ethnic variation with different levels associated with CAD in different populations [12]. Over the last 50 years from when it was first discovered by Norwegian physician Kaare Berg, Lp(a) has evolved from an antigenic determinant in blood type to the strongest genetically determined risk factor for coronary artery disease [12].

The Emerging Risk Factor Collaboration reported a positive association between high Lp (a) levels and cardiovascular, but not all-cause, mortality in a meta-analysis of 24 long-term, prospective studies [13, 14]. Further 2 Danish prospective population studies (Copenhagen City Heart Study and Copenhagen General Population Study) also suggested a possible association between high levels of Lp(a) and all-cause and cardiovascular mortality in the general population. The Danish studies reported that, compared with participants in the bottom 50th percentile of the Lp(a) level distribution (all-cause mortality event rate of 14.2% and cardiovascular mortality event rate of 3.6%), participants with Lp(a) levels above the 95th percentile had

a hazard ratio (HR) for all-cause mortality of 1.20 (95% CI, 1.10–1.30; event rate, 16.5%) and an HR for cardiovascular mortality of 1.50 (95% CI, 1.28–1.76; eventrate5.0%) [15, 16].

The Mendelian Randomisation analysis revealed that Lp(a) levels were associated with an increased long-term risk of all-cause and cardiovascular mortality in 18720 participants of the EPIC-Norfolk prospective population study followed up for a mean of 20 years, in which the mortality risk for those with Lp(a) levels equal to or above the 95th percentile was equivalent to being 1.5 years older in chronologic age [17].

Malignant coronary artery disease (CAD) refers to a severe and extensive atherosclerotic process involving multiple coronary arteries in young individuals (aged <45 years in men and <50 years in women) with a low or no burden of established risk factors. South Asians, in general, develop acute myocardial infarction (AMI) about 10 years earlier and have rates that are three to fivefold higher than in other populations. Although established CAD risk factors have a predictive value, they do not fully account for the excessive burden of CAD in young South Asians. Lp(a) is increasingly recognized as the strongest known genetic risk factor for prematCAD, with high levels observed in South Asians with malignant CAD [18].

The purpose of this analysis was to examine the association of demographic, behavioural and clinical factors including Lp(a) with disease occurrence. The findings from this retrospective analysis would help generate public health evidence for early prediction and risk stratification of CAD, especially in younger adults.

## Methods

### Setting and case definition

Cases included consecutive patients admitted with an Acute Coronary Syndrome (ACS) at the Departments of Cardiology, MAGJ and Lisie hospitals, between 2014–2017. The diagnosis of ACS included an acute ST-Elevation Myocardial Infarction (STEMI), Non-ST Elevation Myocardial Infarction (NSTEMI) and Unstable Angina (UA) as per the Fourth Universal definition of myocardial infarction. Cases also included patients with documented reversible ischaemia on exercise testing, angiographically proven coronary artery disease or a history of re-vascularisation procedures- bypass graft or percutaneous coronary intervention. Obstructive CAD as per the angiogram was defined as a more than 50% obstruction of any one or more of the epicardial coronary arteries. The controls were patients admitted and treated for non-coronary conditions including supraventricular tachycardia (SVT), atrial fibrillation, atypical chest pain with a normal coronary angiogram or who presented as an outpatient for comprehensive health check-ups, with normal results for ischaemia but with one or more identified major risk factors for CAD from diabetes mellitus, hypertension, dyslipidaemia, tobacco use or a family history of CAD. The biochemical analysis included fasting plasma glucose (FBS) and lipid profile. Lp(a) was assessed using the immuno-turbidimetric method using the same test kit (Randox) at both hospitals, FBS by the glucose oxidase method and lipid profile by the CHOD-PAP method. The range of normal for Lp(a) is between 5–30 mg/dl. All tests were done on a HITACH 902 autoanalyzer.

### Data collection

Details of individuals evaluated either as an inpatient or an outpatient were entered on an Excel datasheet which included anthropometric, laboratory parameters, medical conditions, socio-demographic and behavioural factors. Parameters entered included age, sex, body mass index (BMI), waist-hip ratio, disease status vis a vis acute coronary syndrome (ACS), history of diabetes, dyslipidaemia or hypertension, family history of CAD, socio-behavioural habits including tobacco and alcohol use, exercise stress test report, LV function by

echocardiography, coronary angiogram report and revascularization procedure, either percutaneous intervention or coronary bypass graft. Current alcohol user was defined as having had one or more alcoholic drinks over the past 1 year and abstinence as having had no alcoholic drinks over the past 1 year. An Lp(a) value of ≤50mg/dl was considered as normal while values above 50mg/dl were taken as high/elevated.

## Statistical analysis

Bivariate analysis was performed to examine the independent association of the disease with age, sex, diabetes, hypertension, standard BMI categories as defined by the WHO (Underweight (<18.5 Kg/m2); Normal (18.5–24.9 Kg/m2); Overweight and Obese (>= 25 Kg/m$^2$), elevated lipoprotein (>50mg/dl), Dyslipidaemia (LDL>= 130 or HDL<35 or TGL>= 200) tobacco and alcohol use. Continuous variables were presented as mean with standard deviation (SD) and categorical variables were presented as frequencies and percentages. A Chi-square test was used to compare the categorical variables and an independent sample t-test was used to compare the continuous variables. For comparison of median, the Moods median test was used. Multivariable logistic regression analysis was done to investigate the association of demographic, clinical and behavioural factors on the occurrence of disease. All independent factors including Lp(a) that were statistically significant with a p-value of <0.15 in the bivariate analysis were included in the multivariable analysis. Odds ratio with 95% Confidence interval (95% CI) were reported. Two logistic regression models were developed; Model 1 without Lp (a) and Model 2 with Lp(a) to investigate the improvement in the model. A statistically significant (p<0.05) deviance (D = -2logL$_{Model1}$+2logL$_{Model2}$) indicated that the model with more independent variables was a better model to study the association of the factors with the occurrence of the disease. The analysis was performed using SPSS version 20.0.

## Ethics statement

The study was exempted from a review by the IEC, MAGJ hospital since it was a retrospective data analysis using de-identified data from the medical records.

## Results

Data from 928 individuals was included in this study of which 682 had CAD (diseased/cases) and 246 did not have CAD (non-diseased/controls). Of these 169 patients 18.2% had undergone coronary angioplasty of which majority 71.6% were above 50 years of age. The comparison of clinical and demographic characteristics between the cases and controls is shown in Table 1. 67.9% of the cases were above the age of 50 years as compared to 84.3% in the controls. Males were proportionately more in the cases as compared to the controls (75.4% vs 64.5%, p = 0.001). The cases had significantly higher proportion of hypertensives (64.1% vs 32.9, p<0.001), more underweight and normal BMI categories (p = 0.01), more tobacco users (48.7% vs 15.9% p = 0.001) and more alcohol users (29.0% vs 13.4% p<0.001) as compared to the controls. Of those who reported using tobacco (n = 371), 95% (n = 353) were smokers. The mean LDL cholesterol, waist-hip ratio, dyslipidaemia and Lp(a) levels were not significant between the two groups.

Next, we compared the demographic, clinical and behavioural factors between cases and controls in individuals below 50 years. In cases, 39.3% had higher levels of Lp(a) as compared to 19% in controls and this was statistically significant (p = 0.03) In this age group, sex, presence of diabetes, hypertension, tobacco use in any form and alcohol consumption were significantly different between the cases and controls (Table 1).

**Table 1. Comparison of the demographic, clinical and behavioural characteristics between the cases and the controls.**

| Characteristics | All individuals N = 928 | | | Individuals below 50 years N = 186 | | |
|---|---|---|---|---|---|---|
| | Controls N = 246 | Cases N = 682 | p-value | Controls N = 79 | Cases N = 107 | p-value |
| Age group | | | | | | |
| <= 50 years | 79(32.1) | 107(15.7) | 0.0001 | - | - | - |
| >50 years | 167(67.9) | 575(84.3) | | | | |
| Sex | | | | | | |
| Female | 87 (35.4) | 168 (24.6) | 0.0015 | 15 (19.0) | 7 (6.5) | 0.009 |
| Male | 159 (64.6) | 514 (75.4) | | 64 (81.0) | 100 (93.5) | |
| Diabetes | 33 (13.4) | 273 (40.0) | 0.001 | 7 (8.9) | 38 (35.5) | 0.001 |
| Hypertension | 81 (32.9) | 437 (64.1) | 0.001 | 21 (26.6) | 45 (42.1) | 0.03 |
| BMI categories | | | | | | |
| Underweight | 16 (11.9) | 102 (16.7) | | 4 (7.1) | 13 (12.5) | |
| Normal | 80 (59.3) | 399 (65.4) | 0.01 | 32(58.2) | 66 (63.5) | 0.39 |
| Overweight and Obese | 39 (28.9) | 109 (17.9) | | 19 (34.5) | 25 (24.0) | |
| Waist-Hip ratio (Mean ± SD) | 0.92±0.09 | 0.94±0.06 | 0.09 | 0.91±0.14 | 0.93±0.07 | 0.30 |
| Elevated Lp(a) | 71 (28.9) | 218 (32.0) | 0.37 | 15 (19.0) | 42(39.3) | 0.03 |
| LDL Cholesterol (Mean ±S D) | 142.3±50.6 | 139.6±44.9 | 0.44 | 138.5±46.5 | 129.6±39.5 | 0.17 |
| Dyslipidaemia | 68 (28.1) | 217 (32.4) | 0.21 | 56 (72.7) | 76 (71.0) | 0.80 |
| Tobacco use | 39 (15.9) | 332 (48.7) | 0.001 | 16 (20.3) | 64 (59.8) | <0.001 |
| Alcohol consumption | 33 (13.4) | 198 (29.0) | 0.001 | 12 (15.2) | 45 (42.1) | <0.001 |

Figures in parenthesis indicate percentages.

We performed a multivariable logistic regression to examine the relationship between independent variables and the disease status (Table 2). Compared to individuals below 45 years, those in the higher age group were more likely to be diseased. Presence of diabetes (OR = 3.71, 95% CI 2.42–5.70, p<0.01), hypertension (OR = 3.03, 95% CI 2.12–4.34, p<0.01) and tobacco use (OR = 5.44, 95% CI 3.39–8.75, p<0.01) increased the odds of disease. Higher Lp(a) levels increased the odds of the disease by 22% (OR = 1.22, 95% CI 0.87–1.72) but this was not statistically significant. Sex and alcohol use did not increase the likelihood of the disease.

**Table 2. Multivariable logistic regression to study the association of demographic, clinical and behavioural factors.**

| Characteristics | All ages | Below 50 years | |
|---|---|---|---|
| | | Model 1 | Model 2 |
| Age group | | | |
| 46–60 years | 2.55(1.51–4.33)** | | |
| >60 years | 2.54(1.51–4.29)** | | |
| Males | 1.33(0.89–2.0) | 1.44(0.49–4.25) | 1.65(0.53–5.07) |
| Diabetes | 3.71 (2.42–5.70)** | 6.02(2.33–15.58) ** | 7.06(2.59–19.21) ** |
| Hypertension | 3.03(2.12–4.34)** | 1.55(0.73–3.28) | 1.75(0.80–3.84) |
| Elevated LPa | 1.22(0.87–1.72) | | 3.52(1.63–7.57) ** |
| Tobacco use | 5.44(3.39–8.75)** | 5.1(2.22–11.73) ** | 6.38(2.62–15.54) ** |
| Alcohol consumption | 0.96(0.57–1.61) | 1.24(0.49–3.15) | 1.04(0.39–2.76) |

**p<0.01.

For the data from individuals below 50 years, two multivariable logistic regression models were developed; Model 1 regressed all independent factors excluding Lp(a)) with the disease and Model 2 included Lp (a) with other independent factors (Table 2).

Model 1 showed that the presence of diabetes increased the odds of occurrence of the disease by six times (OR = 6.02, 95% CI 2.33–15.58, $p<0.01$) and tobacco use increased the likelihood of disease by 5 times (OR = 5.1, 95% CI 2.22–11.73, $p<0.01$). Being a male, presence of hypertension or consumption of alcohol did not increase the odds of the disease significantly.

Model 2 that included LP(a) showed that diabetes increased the odds of disease by more than seven times (OR = 7.06, 95% CI 2.59–19.21, $p<0.01$) while the use of tobacco increased the odds of disease by more than six times (OR = 6.38, 95% CI 2.62–15.54, $p<0.01$) and Lp(a) increased the odds of disease by more than 3 times (OR = 3.52, 95% CI 1.63–7.57, $p<0.01$). The deviance between Model 1 and Model 2 was 9.82 which was statistically significant at $p<0.05$ when compared with $\chi^2$ with 1 d.f. Thus, Model 2 with Lp(a) is a better logistic regression model for studying the association of the various factors and the disease.

To examine whether Lp(a) varied across the extent of the disease in those below 50 years, we found that the median Lp(a) values increased with severity of the disease and this was statistically significantly (Moods median test, p-value$<0.03$) (Table 3).

## Discussion

This study was conducted on patients with and without coronary artery disease in three tertiary care centres of Kerala in India. Although we studied the association of known factors with the disease viz. Diabetes, hypertension, tobacco and alcohol use, the primary aim was to investigate the effect of raised Lp(a) on the occurrence of the disease. In particular, we investigated the association of Lp(a) as an independent factor in individuals below 50 years. The findings suggest that elevated Lp(a) increased the odds of occurrence of the disease by more than three times in this age group. Sex, BMI, Waist hip ratio and dyslipidemia do not seem to be associated with the disease.

Zampoulakis et al. (2000) studied the relationship of Lp(a) with the extent and severity of atherosclerosis in CAD patients and found that high Lp(a) was associated with more diffuse lesions covering a larger part of the coronary vasculature [19]. Lp(a) levels were also observed to be correlated with the length of coronary lesions as well as the number of diseased vessels especially those with total occlusions [20–23].

A study on 151 patients in South India showed that Lp(a) $>25$ mg/dl increases the risk of CAD by about two-fold [24]. In another study on 71 subjects in North India, Lp(a) concentration of 20–30 mg/dl failed to show a statistically significant association with CAD [7]. Mohan et al. reported a strong association between Lp(a) levels and Intima-Media Thickness (IMT) of the carotid arteries in Type 2 diabetic patients [25]. Angeline et al., tested the Lp(a) levels in 65 patients of myocardial infarction below 45 years old, which when compared to age-matched controls was seen to be significantly higher [26]. Yusuf et al., compared the estimated Lp(a) by immunoturbidimetry in 150 patients each of single-vessel disease (SVD), double vessel disease

**Table 3. Comparison of median Lp(a) with the severity of the disease in individuals below 50 years.**

| Coronary Artery Status | Number of individuals (n = 48) | Median Lp(a) (IQR) |
|---|---|---|
| Normal | 5 | 37.0(12.5–68.0) |
| Single vessel disease | 23 | 43.0(28–68.7) |
| Double vessel disease | 11 | 63.7(22–112.0) |
| Triple vessel disease | 9 | 61(45.8–99.0) |

(DVD) and triple vessel disease (TVD) with a control of 150 healthy volunteers [5]. They found the median Lp(a) levels were significantly raised in cases as compared to controls (median 30.3 vs. 20 mg/dl, p <0.001). Ashfaq et al. [27] reported higher Lp(a) values in 270 patients of CAD as compared to 90 controls without CAD. Lipoprotein (a) 21.0 mg/dL was associated with the presence of coronary lesions ($P$ = 0.0001). Statistically significant differences were seen between patients with normal coronaries as compared with those with SVD, DVD or TVD in this study.

There have been several studies on Lp(a) from various sites in India most of which point towards a positive correlation with atherosclerotic CAD with our series being the largest in India till date comparing the diseased and non-diseased [28, 29]. The baseline values of Lp(a) in our study are higher than most other studies [5, 7, 14, 21]. Importantly, Lp(a) was significantly higher in cases when compared to controls in the age group ≤50 years as also there was a correlation between severity and extent of disease with median Lp(a) values in this age group. In addition, individuals who needed revascularisation, through bypass grafting or percutaneous interventions had higher values than controls. By comparison, in a study by Gupta et al. the mean values of Lp(a) were not different between young myocardial infarctions (MI)< age 35, (Mean Lp(a)37.26 ± 4.06) and old MI (age 36–80) (Mean Lp(a)36.67 ± 4.00) [30]. In another study by Govindaraju et al. in patients of CAD, there were no significant difference in mean Lp(a) values between cases and controls [31]. Lp(a) was studied in 40 patients of myocardial infarction less than age 45 and compared with age-matched controls by Wadhwa et al., who showed that mean Lp(a) in cases (MI < 45 years) was significantly higher as compared with age-matched controls (38.74 +/- 26.15 mg/dl vs 20.54 +/- 16.27 mg/dl) [32]. A similar conclusion was drawn by Bansal et al. [33] from a study in young patients of CAD and the same was corroborated in another study by Ramesh et al. in which Lp(a) was 33.84 ± 23.69 mg/dl in cases (MI age <45 years) and 19.68 ± 10.39 mg/dl in controls [34].

The ESC Guidelines [35] does not recommend plasma Lp(a) for routine risk screening in the general population; however, Lp(a) measurement should be considered in people with high CVD risk or a strong family history of premature atherothrombotic disease. Lp(a) measurement should be considered at least once in each adult person's lifetime to help identify those with very high inherited Lp(a) levels >180 mg/dL (>430 nmol/L) who may have a lifetime risk of ASCVD equivalent to the risk associated with heterozygous familial hypercholesterolaemia [36]. Lp(a) should be considered in selected patients with a family history of premature CVD and for reclassification in people who are borderline between moderate and high-risk [19, 37].

In the light of the current study, we believe Lp(a) to be strongly associated with coronary artery disease in Asian-Indians especially in individuals below fifty years with the disease. Given that Lp(a) values are stable from the first decade of life and considering that Lp(a) values are genetically determined, Lp(a) is found to be strong indicator of premature CAD.

## Limitations of the study

Since the study was carried out at two tertiary care centres in Kerala the results cannot be generalised to the country. The number of diseased was more than 2.5 times the number of non-diseased which does not make for an ideal comparison. The study was carried out in the setting of a hospital in individuals with disease or at risk for disease, the findings of which cannot be generalised to the population. The study is a secondary data analysis that carries all the drawbacks of the same in that the comparator groups are not balanced for common prognostic factors.

## Supporting information

**S1 Data.**
(CSV)

## Acknowledgments

We are thankful for the support received from the departments of Biochemistry, MAGJ Hospital and Lisie hospital.

## Author Contributions

**Conceptualization:** Jacob Joseph, Jaideep Chanayil Menon, Placid K. Sebastien, Abish Sudhakar, Geetha R. Menon.

**Data curation:** Jacob Joseph, Jaideep Chanayil Menon, Abish Sudhakar, Denny John, Geetha R. Menon.

**Formal analysis:** Jaideep Chanayil Menon, Abish Sudhakar, Geetha R. Menon.

**Investigation:** Jacob Joseph.

**Methodology:** Jaideep Chanayil Menon, Placid K. Sebastien.

**Supervision:** Jaideep Chanayil Menon.

**Validation:** Jaideep Chanayil Menon, Placid K. Sebastien.

**Writing – original draft:** Jaideep Chanayil Menon, Denny John, Geetha R. Menon.

**Writing – review & editing:** Jacob Joseph, Jaideep Chanayil Menon, Placid K. Sebastien, Abish Sudhakar, Geetha R. Menon.

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
