## [Decision Letter · Decision Letter 0]

27 Oct 2021

PONE-D-21-20756

Lipoprotein(a) as a risk factor for cardiovascular disease in a South Indian population: A case control study

PLOS ONE

Dear Dr. JAIDEEP CHANAYIL 

Thank you for submitting your manuscript to PLOS ONE. After careful consideration, we feel that it has merit but does not fully meet PLOS ONE’s publication criteria as it currently stands. Therefore, we invite you to submit a revised version of the manuscript that addresses the points raised during the review process.

Please pay close attention to the methods. Address the comments raised by the reviewers and return for second review.

We look forward to receiving your revised manuscript.

Kind regards,

Geofrey Musinguzi, MPH, PhD

Academic Editor

PLOS ONE

“The study was an unfunded one”

3.Thank you for stating the following in your Competing Interests section: 

“None of the authors have any competing interests to declare”

5. Please amend the manuscript submission data (via Edit Submission) to include author Jacob Joseph, Placid K Sebastien, Abish Sudhakar, Denny John, Geetha R Menon.

Reviewers' comments:

Reviewer's Responses to Questions

**Comments to the Author**

1. Is the manuscript technically sound, and do the data support the conclusions?

Reviewer #1: Yes

Reviewer #2: Partly

2. Has the statistical analysis been performed appropriately and rigorously? 

Reviewer #1: No

Reviewer #2: No

3. Have the authors made all data underlying the findings in their manuscript fully available?

Reviewer #1: Yes

Reviewer #2: No

4. Is the manuscript presented in an intelligible fashion and written in standard English?

Reviewer #1: No

Reviewer #2: Yes

5. Review Comments to the Author

Reviewer #1: PONE-D-21-20756

General comments:

1. This is a good study question and I would encourage the authors to make the presentation better.

2. I believe that South Asians is better than Asian Indians. Most of the global studies use data from South Asia and not India alone.

3. The basic limitation in the study is use of mean levels of lipoprotein(a). Lp(a) typically has a skewed distribution and I would recommend that the authors use median values with 25-75 interquartile intervals (IQR) or data presentation. Similarly intergroup comparisons should be performed with non-parametric statistics (Kruskall-Wallis or other tests). An expert opinion from a statistician should be obtained. In the current form the data are not understandable, SD’s are more than the mean values. I am skeptical for use of other parametric tests in the statistics.

4. The article is too lengthy. Especially the introduction and discussion sections.

Specific comments:

Abstract:

5. The Background statement is lengthy. Please make it a one liner.

6. Please describe the method of identifying controls here.

7. All the results could change after the change in statistical methodology highlighted in point no. 3 above.

Introduction:

8. Please describe the evolution of Lp(a) as a risk factor especially focusing on Emerging Risk Factors Collaboration, Danish studies and Mendelian Randomization studies. To be identified it is essential that the factor follows all the Bradford-Hill criteria.

9. Remove the redundant and duplicate statements from this section. Paragraphs 1, 3, 4, 5, 6, and 7 could be deleted/shortened. Previous studies should be could be referenced and details could be discussed in Discussion. No need to provide details of Lp(a) structure.

10. I agree that this is a large study. However, this is not the largest study, INTERHEART study Lp(a) data have been published with more cases/controls than the present one.

11. The literature review is incomplete.

12. Objectives are not defined at all.

Methods:

13. Ethics clearance should be in the first paragraph.

14. Too many abbreviations make for a tough reading. The language needs major improvement so as to improve understanding.

15. I would not include individuals with less than 50% obstructive lesions as controls. These are the persons in whom acute coronary events happen, especially in the young and should be included in CAD group.

16. Most of the guidelines, including latest European and North American, recommend a cut-off of >50 mg/dl and high Lp(a). This cut off should be used. >30mg/dl could be a secondary cut-off.

17. Statistical analyses need major revision. Highlighted in point no. 3 supra. Why adjust for age, when age is the defining characteristics and age-stratified analyses have ben done?

Results:

18. Please describe the characteristics of non-CAD patients.

19. In Table 1, there are too many data. Anyway, this should be revised after exclusion of controls with any evidence of CAD on coronary angio.

20. Dyslipidemia is a wrong word. Please specify the hyperlipidemias as, LDL >=130, non-HDL >=160, and TG >=150. Similarly low HDL should be better defined according to international guidelines. Arbitrary cut-offs are not recommended.

21. Alcohol intake Y/N is also wrong. Please specify the alcohol intake cut-off used in international studies.

22. Age-stratified data on Table 2 should be reduced or combined in Table 1.

23. Presentation of statistical analyses are not understandable at all. The problem with the currently available statistical packages is that they produce too many outputs which are not relevant to the study question. Please take guidance from an expert to create more useful statistics.

24. Figure 1 should be a bar graph and not a line graph.

25. Discussion:

26. The first paragraph should report principal conclusions of the study.

27. Too much space has been taken up by pathophysiology of Lp(a). This must be deleted.

28. Review of Indian studies is incomplete. Also include some studies from other South Asian countries.

29. The current guidelines recommend measurement of Lp(a) in every individual, at least once in lifetime. In Indian context it could be more than once, given the varies of laboratory variations.

30. Limitations should be more specific. This is not a prospective study, sample size is small (although larger than most Indian studies), control population could be better selected as age-matching is uneven (more young patients than controls), etc.

31. Too much repetition.

32. The length of the Discussion section should be less.

Reviewer #2: Manuscript Number: PONE-D-21-20756

Lipoprotein(a) as a risk factor for cardiovascular disease in a South Indian population: A case control study

PLOS ONE

Thank you for the opportunity to review this manuscript. The manuscript identified lipoprotein(a) as a risk factor (predictor?) for CVD, particularly those aged below 50 years. Overall, the manuscript is good in its concept and will be of interest to those involved in CVD prevention and management. However, I have major issues in the novelty of the study, methodological issues and results that need to be addressed critically. Specific comments and questions are provided below.

General

1. The present manuscript reported on a topic that has been studied previously in India as well as other parts of the world. Many studies, both predictive and causal, have already been published on this topic and found somewhat similar results, indicating LP (a) is an independent risk factor of CAD. On paragraph 6 of the background, several studies have been mentioned from India, which showed more or less consistent results. However, with the abundance of such studies, the motivation to conduct this study is not well justified. What makes this study different from the others in terms of study population, methodology, approach, or context other than a bit larger sample? The authors need to explain why their study was necessary, and what it adds to common knowledge.

2. It is essential to clearly state whether this is a ‘prediction’ study that identifies predictors of CAD or ‘causal’ which needs controlling for confounding and adjustment for selection biases. Based on the description from the introduction and method section, the aim of this study looks more of causal, i.e., means to identify the role of LP (a) as a risk for CAD. However, in the analysis and result section, the authors described the ROC curve, which is completely prediction (discriminatory potential) of LP(a) for CAD, means…confounding is not an issue (Grobbee and Hoes 2014). Moreover, in paragraph 4 of the discussion, the authors emphasize LP(a) to be considered to reclassify patients, which is completely prediction in nature. Thus, the authors should explicitly describe whether the aim is prediction or causal and descriptions should be consistent throughout the manuscript. In addition, if the aim is causal, there are several biochemical parameters (lipid profile) that need to be controlled to claim that LP(a) is an independent risk factor of CAD.

Title

3. The title refers to cardiovascular diseases, which includes several categories, which is not limited to coronary artery diseases (CAD). However, the study is specifically for CAD. There are several risk factors which are limited to specific CVD types. If the outcome is CVD, controls are also CVD patients other than CAD. Thus, I would suggest limiting the title specific to CAD, which is the focus of this study.

Background

4. Introduction, paragraph 4: the first two descriptions do not have references.

5. The role of paragraph 7 of the background is not clear or not well placed in the background.

Methods

6. How the number of cases and control (the ratio) was determined? It is an unbalanced case to control ratio.

7. Controls were supraventricular tachycardia (SVT), atrial fibrillation, atypical chest pain with a normal coronary angiogram. First, these cardiac conditions might be related to the main exposure of interest, i.e. LP(a). This may underestimate the association between LP(a) and CAD. The study would have benefited from multiple control groups. Would it have been better to compare the cases with healthy counterparts in addition to these cases?

8. Ethics statement: It is not clear how this study is exempted from ethical review. It is acceptable that consent might not be required as it is a hospital record. However, it doesn’t necessarily mean ethical review is not required.

Analysis

9. In the analysis section, the authors described that Lp(a) is not normally distributed. However, in the result section of the abstract and the main document, LP(a) is described using mean (SD). This is evidenced by table 4 of the result section, which indicates huge variation in mean and median. The authors described that they used log-transformation. Nevertheless, throughout the manuscript, raw values of LP(a) are reported. Thus, the authors should revise the analysis and results accordingly.

10. Receiver operating characteristic curve (ROC) analysis was used to obtain the predictive accuracy of Lp(a) by disease status. What is the importance of ROC…if the study aim is causal.

11. LP(a) is continuous variable and treated as continuous in one occasion and as categorical in another place. It is better to make it consistent based on the prior hypothesis.

12. Table 1 and 2: ‘lipoprotein’, which specific type of lipoprotein?

13. It is puzzling that the main exposure of interest (Lipoprotein) is not included in the multivariable analysis for ‘all ages’.

Results

14. The authors described that “We found that Lp(a) was able to prediction accuracy of Lp(a) was 57% as compared to 62% for diabetes and 70% for tobacco use (Table 5). The ROC curve drawn with these variables in the model gives a predictive probability of about 80%”. This indicates that the predictive capacity of LP(a) is very low…only 7% over just tossing a coin. It would be better to see the added value of LP(a) apart from other predictors including, tobacco, diabetes, BP, BMI, etc.…

Discussion

15. Last paragraph of the discussion section: “Given that Lp(a) values are stable from the first decade of life and considering that Lp(a) values are genetically determined, Lp(a) should be considered for risk stratification in families with premature CAD and or with high Lp(a) levels in the family”. Based on this description, the aim of this study is prognostication in its nature, i.e. to evaluate the prognostic performance of LP(a) in predicting CAD. If so, the AUC is only 57%, which is very low, a bit higher than chance. If the aim is not to see the causal role of LP(a) so as to guide preventive interventions, I would suggest restructuring the whole manuscript as a prognostic research. The theoretical design and approach to statistical analysis is different for causal and prediction research.

Minor issues

• In general, the manuscript doesn’t have page and line numbers, which make the review process difficult to refer to specific descriptions.

• Be consistent in the use of decimal points

References

Grobbee, D. E. and A. W. Hoes (2014). Clinical epidemiology: principles, methods, and applications for clinical research, Jones & Bartlett Publishers.

6. PLOS authors have the option to publish the peer review history of their article (what does this mean?). If published, this will include your full peer review and any attached files.

Reviewer #1: No

Reviewer #2: **Yes: **Hamid Yimam Hassen

---

## [Author Response · Author response to Decision Letter 0]

16 Dec 2021

I thought both the reviewers comments were excellent and helped us to give the manuscript a better form. 

Reviewer #1: PONE-D-21-20756

General comments:

Comment 1. This is a good study question and I would encourage the authors to make the presentation better.

Response: We thank the reviewer for appreciating the idea. We have incorporated each of the comments made by the reviewer in the revised manuscript.

Comment 2. I believe that South Asians is better than Asian Indians. Most of the global studies use data from South Asia and not India alone.

Response: Asian-Indians has been replaced by South Asians as suggested

Comment 3. The basic limitation in the study is use of mean levels of lipoprotein(a). Lp(a) typically has a skewed distribution and I would recommend that the authors use median values with 25-75 interquartile intervals (IQR) or data presentation. Similarly intergroup comparisons should be performed with non-parametric statistics (Kruskall-Wallis or other tests). An expert opinion from a statistician should be obtained. In the current form the data are not understandable, SD’s are more than the mean values. I am skeptical for use of other parametric tests in the statistics.

Response. We thank the reviewer for the observations. The Lp(a) median values (IQR) has been compared between the disease categories using Mood’ s median test. This has been presented in Table 3 in the revised manuscript.

Comment 4. The article is too lengthy. Especially the introduction and discussion sections.

Response The introduction and the discussion section has been shortened considerably. 

Specific comments:

Abstract:

Comment 5. The Background statement is lengthy. Please make it a one liner.

Response: The background statement has been shortened to a single sentence as suggested

Comment 6. Please describe the method of identifying controls here.

Response: As mentioned in the methods section controls were patients admitted and treated for non-coronary conditions including supraventricular tachycardia (SVT), atrial fibrillation, atypical chest pain with a normal coronary angiogram or who presented as an outpatient for comprehensive health check-ups, with normal results for ischaemia but with one or more identified major risk factors for CAD including diabetes mellitus, hypertension, dyslipidaemia, tobacco use or a family history of CAD. 

Comment 7. All the results could change after the change in statistical methodology highlighted in point no. 3 above.

Response: The results are incorporated in the revised manuscript using Median and IQR as suggested.

Introduction:

8. Please describe the evolution of Lp(a) as a risk factor especially focusing on Emerging Risk Factors Collaboration, Danish studies and Mendelian Randomization studies. To be identified it is essential that the factor follows all the Bradford-Hill criteria.

Response: The mentioned studies have been incorporated into the Introduction section.

9. Remove the redundant and duplicate statements from this section. Paragraphs 1, 3, 4, 5, 6, and 7 could be deleted/shortened. Previous studies should be could be referenced and details could be discussed in Discussion. No need to provide details of Lp(a) structure.

Response: Changes as suggested have been made and the section on Lp(a) structure has been deleted in its entirety. 

10. I agree that this is a large study. However, this is not the largest study, INTERHEART study Lp(a) data have been published with more cases/controls than the present one

Response: The total number of subjects in whom Lp(a) was tested as a part of the INTERHEART study from South Asia (India, Pakistan, Bangladesh and Nepal was 1818, which most likely makes the present study probably larger than the INTERHEART as far as sampling for Lp(a) were concerned. 

11. The literature review is incomplete.

Response: The mentioned trials from Denmark, the Risk factor Collaboration and the Mendelian Randomisation have been added. 

12. Objectives are not defined at all.

Response: The last paragraph of the Introduction defines the Objectives and has been labelled so in the revised version.

Methods:

13. Ethics clearance should be in the first paragraph.

Response: Ethics clearance has been now included as the first paragraph as suggested.

14. Too many abbreviations make for a tough reading. The language needs major improvement so as to improve understanding.

Response: We have modified the manuscript to make for a better read.

15. I would not include individuals with less than 50% obstructive lesions as controls. These are the persons in whom acute coronary events happen, especially in the young and should be included in CAD group.

Response: Individuals with minor CAD have been included in the patient arm as suggested while those with normal epicardial coronaries were in the control arm.

16. Most of the guidelines, including latest European and North American, recommend a cut-off of >50 mg/dl and high Lp(a). This cut off should be used. >30mg/dl could be a secondary cut-off.

Response: Modification to the manuscript have been made taking a cut-off of > 50 mg/dl to denote high Lp(a) as suggested. 

17. Statistical analyses need major revision. Highlighted in point no. 3 supra. Why adjust for age, when age is the defining characteristics and age-stratified analyses have been done?

Results: The reviewers point on statistical analysis has been taken. The binary logistic regression on all ages has age as a predictor variable and has been adjusted for age. However the logistic regression analysis on persons below the age of 50 years is not adjusted for age.

18. Please describe the characteristics of non-CAD patients.

Response: The characteristics of non-CAD patients (non-diseased) is described in table 1.

19. In Table 1, there are too many data. Anyway, this should be revised after exclusion of controls with any evidence of CAD on coronary angio.

Response: Table 1 and 2 have been combined into one table giving the patient characteristics.

20. Dyslipidemia is a wrong word. Please specify the hyperlipidemias as, LDL >=130, non-HDL >=160, and TG >=150. Similarly low HDL should be better defined according to international guidelines. Arbitrary cut-offs are not recommended.

Response: The suggested modifications have been made

21. Alcohol intake Y/N is also wrong. Please specify the alcohol intake cut-off used in international studies.

Response: The intake cut-off used are as per international studies and has been re-written. Current drinking was defined as having had one or more alcoholic drinks over the past 1 year and abstinence as having had no alcoholic drinks over the past 1 year

22. Age-stratified data on Table 2 should be reduced or combined in Table 1.

Response: Table 1 and 2 have now been combined into a single table.

23. Presentation of statistical analyses are not understandable at all. The problem with the currently available statistical packages is that they produce too many outputs which are not relevant to the study question. Please take guidance from an expert to create more useful statistics.

Response: The authors have interpreted the logistic regression analysis in a way that is easily understood by the clinicians. This is the standard way of reporting the results. 

24. Figure 1 should be a bar graph and not a line graph.

Response: We have removed the figure in the revised manuscript 

25. Discussion:

26. The first paragraph should report principal conclusions of the study.

Response: Principal conclusions of the study have been included in the first paragraph as suggested.

27. Too much space has been taken up by pathophysiology of Lp(a). This must be deleted.

Response: Pathophysiology has been deleted.

28. Review of Indian studies is incomplete. Also include some studies from other South Asian countries.

Response: Not aware of any other Indian studies.

29. The current guidelines recommend measurement of Lp(a) in every individual, at least once in lifetime. In Indian context it could be more than once, given the varies of laboratory variations.

Response: Agree with the reviewer.

30. Limitations should be more specific. This is not a prospective study, sample size is small (although larger than most Indian studies), control population could be better selected as age-matching is uneven (more young patients than controls), etc.

Response: The above mentioned have been incorporated into the revised manuscript.

31. Too much repetition.

Response: Necessary deletions made.

32. The length of the Discussion section should be less.

Response: Discussion has been shortened.

Reviewer #2: Manuscript Number: PONE-D-21-20756

Lipoprotein(a) as a risk factor for cardiovascular disease in a South Indian population: A case control study

PLOS ONE

Thank you for the opportunity to review this manuscript. The manuscript identified lipoprotein(a) as a risk factor (predictor?) for CVD, particularly those aged below 50 years. Overall, the manuscript is good in its concept and will be of interest to those involved in CVD prevention and management. However, I have major issues in the novelty of the study, methodological issues and results that need to be addressed critically. Specific comments and questions are provided below.

General

1. The present manuscript reported on a topic that has been studied previously in India as well as other parts of the world. Many studies, both predictive and causal, have already been published on this topic and found somewhat similar results, indicating LP (a) is an independent risk factor of CAD. On paragraph 6 of the background, several studies have been mentioned from India, which showed more or less consistent results. However, with the abundance of such studies, the motivation to conduct this study is not well justified. What makes this study different from the others in terms of study population, methodology, approach, or context other than a bit larger sample? The authors need to explain why their study was necessary, and what it adds to common knowledge.

Response: Thanks for your review and the clarity you have brought into the review;

Other than for being the largest in India by number, we have compared Lp(a) levels between patients and controls in the ≤50 years age and compared Lp(a) with other risk factors for disease as predictor for CAD in that age group. This because a number of the younger age patients with coronary heart disease (CHD) do not have any of the conventional risk factors for CHD. Our study was an attempt to assess as to how predictive Lp(a) is especially in the age group < 50 years. 

2. It is essential to clearly state whether this is a ‘prediction’ study that identifies predictors of CAD or ‘causal’ which needs controlling for confounding and adjustment for selection biases. Based on the description from the introduction and method section, the aim of this study looks more of causal, i.e., means to identify the role of LP (a) as a risk for CAD. However, in the analysis and result section, the authors described the ROC curve, which is completely prediction (discriminatory potential) of LP(a) for CAD, means…confounding is not an issue (Grobbee and Hoes 2014). Moreover, in paragraph 4 of the discussion, the authors emphasize LP(a) to be considered to reclassify patients, which is completely prediction in nature. Thus, the authors should explicitly describe whether the aim is prediction or causal and descriptions should be consistent throughout the manuscript. In addition, if the aim is causal, there are several biochemical parameters (lipid profile) that need to be controlled to claim that LP(a) is an independent risk factor of CAD.

Response: The study is more predictive than causal and in the revised version we have addressed the concerns stated.

Title

3. The title refers to cardiovascular diseases, which includes several categories, which is not limited to coronary artery diseases (CAD). However, the study is specifically for CAD. There are several risk factors which are limited to specific CVD types. If the outcome is CVD, controls are also CVD patients other than CAD. Thus, I would suggest limiting the title specific to CAD, which is the focus of this study.

Response: Thank you, we have changed the title as suggested to coronary artery disease (CAD). 

Background

4. Introduction, paragraph 4: the first two descriptions do not have references.

Response: The reference for the section on “malignant CAD” is 18- Enas K Enas et al. 

5. The role of paragraph 7 of the background is not clear or not well placed in the background.

Response: Lp(a) is not routinely assessed in patients with an ACS in most parts of India due to multiple reasons one of which being the lack of any treatment for lowering Lp(a). There are a number of molecules currently being tested which have shown benefit in lowering Lp(a) and hence this paragraph was incorporated.

It has been removed and is placed in the Discussion section

Methods

6. How the number of cases and control (the ratio) was determined? It is an unbalanced case to control ratio.

Response: The cases and controls comprised of patients of CAD as defined in the text as cases to controls of patients admitted and treated for non-ischaemic cardiac conditions or in those who for whom it was done as a part of a comprehensive health check-up provided that the individual had one or more of the 5 risk factors for CAD- diabetes, HTN, dyslipidaemia, tobacco use or a family history of CAD.

7. Controls were supraventricular tachycardia (SVT), atrial fibrillation, atypical chest pain with a normal coronary angiogram. First, these cardiac conditions might be related to the main exposure of interest, i.e. LP(a). This may underestimate the association between LP(a) and CAD. The study would have benefited from multiple control groups. Would it have been better to compare the cases with healthy counterparts in addition to these cases?

Response: That is a very good suggestion which we hope we will be able to take up in the future. 

8. Ethics statement: It is not clear how this study is exempted from ethical review. It is acceptable that consent might not be required as it is a hospital record. However, it doesn’t necessarily mean ethical review is not required.

Response: The study was exempted from a ethical review process since it was a secondary data analysis. 

Analysis

9. In the analysis section, the authors described that Lp(a) is not normally distributed. However, in the result section of the abstract and the main document, LP(a) is described using mean (SD). This is evidenced by table 4 of the result section, which indicates huge variation in mean and median. The authors described that they used log-transformation. Nevertheless, throughout the manuscript, raw values of LP(a) are reported. Thus, the authors should revise the analysis and results accordingly.

Response: Thanks for the suggestion which has been taken and necessary changes made.

10. Receiver operating characteristic curve (ROC) analysis was used to obtain the predictive accuracy of Lp(a) by disease status. What is the importance of ROC…if the study aim is causal.

Response: The study looks at the predictive accuracy of Lp(a). The ROC analysis was done to determine the predictive accuracy of Lp(a) as compared to other established risk factors for CAD.

11. LP(a) is continuous variable and treated as continuous in one occasion and as categorical in another place. It is better to make it consistent based on the prior hypothesis.

Response: It is now primarily treated as a categorical variable.

12. Table 1 and 2: ‘lipoprotein’, which specific type of lipoprotein?

Response: Thanks, has been labelled as Lp(a)

13. It is puzzling that the main exposure of interest (Lipoprotein) is not included in the multivariable analysis for ‘all ages’.

Response: Lp(a) was not included in the multiple logistic regression because it did not show statistical significance between the diseased and non-diseased groups in the univariate analysis, however in < 50 age group it was included in the regression analysis as it was significant in the univariate analysis.

Results

14. The authors described that “We found that Lp(a) was able to prediction accuracy of Lp(a) was 57% as compared to 62% for diabetes and 70% for tobacco use (Table 15). The ROC curve drawn with these variables in the model gives a predictive probability of about 80%”. This indicates that the predictive capacity of LP(a) is very low…only 7% over just tossing a coin. It would be better to see the added value of LP(a) apart from other predictors including, tobacco, diabetes, BP, BMI, etc.…

Response: The authors appreciate the remarks of the reviewer, the ROC analysis with a cut-off of 50mg% as high Lp(a) yields a predictive accuracy of 60% which is statistically significant. This is almost as good as diabetes which has an accuracy of 62%. The logistic regression analysis in ≤50 years shows the added value of Lp(a) with the other known predictors.

Discussion

15. Last paragraph of the discussion section: “Given that Lp(a) values are stable from the first decade of life and considering that Lp(a) values are genetically determined, Lp(a) should be considered for risk stratification in families with premature CAD and or with high Lp(a) levels in the family”. Based on this description, the aim of this study is prognostication in its nature, i.e. to evaluate the prognostic performance of LP(a) in predicting CAD. If so, the AUC is only 57%, which is very low, a bit higher than chance. If the aim is not to see the causal role of LP(a) so as to guide preventive interventions, I would suggest restructuring the whole manuscript as a prognostic research. The theoretical design and approach to statistical analysis is different for causal and prediction research.

Response: Thanks, the manuscript has been restructured towards predictive research as suggested.

Minor issues

• In general, the manuscript doesn’t have page and line numbers, which make the review process difficult to refer to specific descriptions.

• Be consistent in the use of decimal points

Has been rectified

References

Grobbee, D. E. and A. W. Hoes (2014). Clinical epidemiology: principles, methods, and applications for clinical research, Jones & Bartlett Publishers.

6. PLOS authors have the option to publish the peer review history of their article (what does this mean?). If published, this will include your full peer review and any attached files.

Do you want your identity to be public for this peer review? For information about this choice, including consent withdrawal, please see our Privacy Policy.

Reviewer #1: No

Reviewer #2: Yes: Hamid Yimam Hassen

---

## [Decision Letter · Decision Letter 1]

26 Jan 2022

PONE-D-21-20756R1Lipoprotein(a) as a risk factor for coronary artery disease in a South Asian population: A case control studyPLOS ONE

Dear Dr. JAIDEEP CHANAYIL MENON,

Thank you for submitting your manuscript to PLOS ONE. After careful consideration, we feel that it has merit but does not fully meet PLOS ONE’s publication criteria as it currently stands. Therefore, we invite you to submit a revised version of the manuscript that addresses the points raised during the review process.

Both reviewers still have some important methodological concerns that I advise that you address and return the manuscript for consideration.

We look forward to receiving your revised manuscript.

Kind regards,

Geofrey Musinguzi, MPH, PhD

Academic Editor

PLOS ONE

Journal Requirements:

Reviewers' comments:

Reviewer's Responses to Questions

**Comments to the Author**

1. If the authors have adequately addressed your comments raised in a previous round of review and you feel that this manuscript is now acceptable for publication, you may indicate that here to bypass the “Comments to the Author” section, enter your conflict of interest statement in the “Confidential to Editor” section, and submit your "Accept" recommendation.

Reviewer #1: All comments have been addressed

Reviewer #2: (No Response)

2. Is the manuscript technically sound, and do the data support the conclusions?

Reviewer #1: Partly

Reviewer #2: Partly

3. Has the statistical analysis been performed appropriately and rigorously? 

Reviewer #1: No

Reviewer #2: No

4. Have the authors made all data underlying the findings in their manuscript fully available?

Reviewer #1: Yes

Reviewer #2: Yes

5. Is the manuscript presented in an intelligible fashion and written in standard English?

Reviewer #1: Yes

Reviewer #2: Yes

6. Review Comments to the Author

Reviewer #1: Thanks for submitting a revised version of this article. The manuscript is now much better, but some deficiencies remain.

1. The authors have log transformed the Lp(a) values and then performed the quantitative analyses. This is also an appropriate method, although I believe that using non-parametric comparisons such as median test for 2 group and Kruskal-Wallis test for multiple group comparisons are better methods. This should be discussed in the limitation sections of the article.

2. The interquartile ranges are typically presented as median (25-75 interquartile range, IQR), e.g. 55 (40-70). Nowhere the authors have presented data in this format. Moreover, the authors have reported mean levels of Lp(a) without any measurement of dispersion in the abstract, text and tables. All these should be revised.

3. The discussion section is still too long and instea dof discussing the findings of the article and comparing these with previous Indian and international studies, the authors focus on implications of the findings. Please modify.

4. Studies on Lp(a) from India are listed below. There are many such studies and should be referenced.

Table: Summary of case-control studies regarding importance of Lp(a) in CAD in India

First Author Year Journal Cases Lp(a) (n) Controls Lp(a) (n) Comment

Vashisht S 1992 Indian Heart J. 44(4):223-6 Lp(a)+++ (760)** Lp(a)+ (167)** Clinical CAD

Gupta R 1996 Int J Cardiol. 57(3):265-70 26.8+22.1 (77) 15.1+14.6 (24) Angiographic study

Mohan V 1998 Diabetes Care. 21(11):1819-23 24.6 (100)* 15.1 (100), 19.4 (100)* Diabetic CAD

Singh S 1999 J Assoc Phys Ind. 47(12):1157-60 (222) (67) Post MI

Gupta R 2000 Indian Heart J. 52(4):407-10 11.9+2.8 (48)# 6.7+3.4 (23)# Recent CAD <60y

Gambhir JK 2000 Indian Heart J. 52(4):511-5 35.0+32.4 (50) 20.3+17.0 (50) Young CAD <40y

Govindraju V 2003 J Ind Med Assoc. 101(8):458-60 32.2+1.4 (300) 30.0+2.6 (200) Stable CAD

Rajasekhar D 2004 Ind J Clin Biochem. 19(2):53-9 24.8+19.0 (151) 16.0+17.5 (49) Stable CAD

Wadhwa A 2013 J Assoc Phys Ind. 61(6):384-6 38.7+26.2 (40) 26.2+20.5 (40) Young MI <45y

Yusuf J 2014 Indian Heart J. 66(3):272-9 30.3 (450)* 20.0 (150)*

Bansal SK 2015 J Clin Diagn Res. 9(11):BC07-11 43.2+10.2 (30) 17.6+3.2 (30) Stable CAD <60y

Ramesh G 2018 Interv Med Appl Sci. 10(2):65-9 33.8+23.7 (51) 19.7+10.4 (51) Young MI <45y

Gupta MD 2018 Ind Heart J. 70(S3):S146-56 37.3+4.1 (125) 27.3+3.4 (103) Young MI <35y

Hanif S 2019 Pak J Med Sci. 35(6):1718-23 47.0+45.5 (90) 29.7+23.1 (90) Young MI <45y

*median value; #geometric mean; CAD coronary artery disease; MI myocardial infarction; **immunofluorescence technique

Reviewer #2: Manuscript Number: PONE-D-21-20756R1

Lipoprotein(a) as a risk factor for coronary artery disease in a South Asian population: A case control study

PLOS ONE

Thank you again for the opportunity to review this manuscript. However, I still have major issues that need to be addressed critically. Specific comments and questions are provided below.

1. The authors claim that it is a prediction study not causal or risk factor analysis. However, the title still reads as “…as a risk factor….”. The theoretical design, the approach to data analysis and interpretation of the study varies significantly for prediction and causal research. The description in the abstract and the whole body of the manuscript still reads as ‘…risk factor..’. In prediction, individual ORs do not have interpretive value. Prediction is not necessarily risk factor identification.

2. I still suggest authors consider evaluating the added value of LP(a) in predicting CAD along with other predictors such as tobacco, DM, etc.…. Whether statistically significant or not, AUC of 60% has minimal clinical importance. It is better to develop a multivariable prediction model with and without LP(a) and assess the added value.

3. Although LP(a) is not normally distributed, using mean with SD is not justified enough.

4. Whether significant or not on the univariable analysis, the main predictor of interest (LP (a)) should be included in the multivariable analysis.

5. I commend the authors for uploading the data used to produce this manuscript. It helps for open science. Looking in to the supporting information (Data), there are lots of missing values. However, there is no description on the missing data. Did the authors check for the missingness pattern? How did they manage it?

6. In addition, the outcome variable and predictors are not clearly labeled in the data. I suggest authors make it more clear and transparent.

• In spite of the comment in round 1 to add for line numbers, the manuscript still doesn’t have page and line numbers, which make the review process difficult to refer to specific descriptions. As far as I understand, the author guideline requests to put line numbers to ease the review process.

7. PLOS authors have the option to publish the peer review history of their article (what does this mean?). If published, this will include your full peer review and any attached files.

Reviewer #1: No

Reviewer #2: **Yes: **Hamid Y. Hassen

---

## [Author Response · Author response to Decision Letter 1]

4 Feb 2022

Reviewer #1: Thanks for submitting a revised version of this article. The manuscript is now much better, but some deficiencies remain.

Response: We thank the reviewer for his encouraging remarks.

Comment 1. The authors have log transformed the Lp(a) values and then performed the quantitative analyses. This is also an appropriate method, although I believe that using non-parametric comparisons such as median test for 2 group and Kruskal-Wallis test for multiple group comparisons are better methods. This should be discussed in the limitation sections of the article.

Response: We have reported the median Lp(a) values and used the median test for comparison in the revised version of the manuscript. (Line 23-26, page 3 and Table 4 page 11) 

Comment 2. The interquartile ranges are typically presented as median (25-75 interquartile range, IQR), e.g. 55 (40-70). Nowhere the authors have presented data in this format. Moreover, the authors have reported mean levels of Lp(a) without any measurement of dispersion in the abstract, text and tables. All these should be revised.

Response: We have reported the (25-75 IQR) in table 4 page 11 of the revised manuscript.

Comment 3. The discussion section is still too long and instead of discussing the findings of the article and comparing these with previous Indian and international studies, the authors focus on implications of the findings. Please modify.

Response: The discussion section has been suitably altered comparing between previous studies and the present one. The implications of our study have been removed from the section. 

4. Studies on Lp(a) from India are listed below. There are many such studies and should be referenced.

Response: Thank you very much, all the references have been included and cited.

Reviewer #2: Manuscript Number: PONE-D-21-20756R1

Lipoprotein(a) as a risk factor for coronary artery disease in a South Asian population: A case control study

PLOS ONE

Thank you again for the opportunity to review this manuscript. However, I still have major issues that need to be addressed critically. Specific comments and questions are provided below.

Response: We thank the reviewer for his comments. We have responded to each one of his comments in the revised manuscript

Comment 1. The authors claim that it is a prediction study not causal or risk factor analysis. However, the title still reads as “…as a risk factor….”. The theoretical design, the approach to data analysis and interpretation of the study varies significantly for prediction and causal research. The description in the abstract and the whole body of the manuscript still reads as ‘…risk factor.’. In prediction, individual ORs do not have interpretive value. Prediction is not necessarily risk factor identification.

Response: Thanks for the detailed observation. We have studied the association of Lp(a) with CAD in the presence of other known factors. We have changed the title of the manuscript now. 

Comment 2. I still suggest authors consider evaluating the added value of LP(a) in predicting CAD along with other predictors such as tobacco, DM, etc.…. Whether statistically significant or not, AUC of 60% has minimal clinical importance. It is better to develop a multivariable prediction model with and without LP(a) and assess the added value.

Response: We thank the reviewer for his valuable suggestions. We have now added another table for logistic regression for individuals below 50 years(Table 3) in the revised manuscript in page 10. Two models Model 1 and Model 2 have been developed. Model 1 has CAD has the dependent variable and gender(female/male), diabetes(yes/no), hypertension(yes/no), tobacco use (yes/no), alcohol use (yes/no) as predictor variables and Model 2 has CAD has the dependent variable and gender(female/male), diabetes(yes/no), hypertension(yes/no), tobacco use (yes/no), alcohol use (yes/no) and Lp(a)(<50 and >=50) as predictors. The deviance between Model 1 and Model 2 was 9.82 which is statistically significant at p<0.05 when compared with �2 with 1 d.f. Thus, Model 2 with Lp(a) is a better predictor model for the disease as compared to Model 1 without it, indicating that Lp(a) is an independent predictor for the disease in addition to the established risk factors viz. presence of diabetes and tobacco use

Comment 3. Although LP(a) is not normally distributed, using mean with SD is not justified enough.

Response: We have not reported mean Lp(a) anywhere in the revised manuscript now.

Comment 4. Whether significant or not on the univariable analysis, the main predictor of interest (LP (a)) should be included in the multivariable analysis.

Response: We have now revised Table 1 for all individuals after including Lp(a) as a predictor in the multivariable analysis.

Comment 5. I commend the authors for uploading the data used to produce this manuscript. It helps for open science. Looking in to the supporting information (Data), there are lots of missing values. However, there is no description on the missing data. Did the authors check for the missingness pattern? How did they manage it?

Response: We had uploaded a revised data sheet with the variables used for the analysis.

Comment6. In addition, the outcome variable and predictors are not clearly labeled in the data. I suggest authors make it more clear and transparent.

Response: A revised data sheet is being uploaded with clear labels

Comment 7. In spite of the comment in round 1 to add for line numbers, the manuscript still doesn’t have page and line numbers, which make the review process difficult to refer to specific descriptions. As far as I understand, the author guideline requests to put line numbers to ease the review process.

Response: Thanks for the suggestions. We have now inserted the page numbers and the line numbers now in the revised manuscript.

---

## [Decision Letter · Decision Letter 2]

8 Mar 2022

PONE-D-21-20756R2Association of lipoprotein(a) with coronary artery disease in a South Asian population: A case control studyPLOS ONE

Dear Dr. Menon,

Thank you for submitting your manuscript to PLOS ONE. After careful consideration, we feel that it has merit but does not fully meet PLOS ONE’s publication criteria as it currently stands. Therefore, we invite you to submit a revised version of the manuscript that addresses the points raised during the review process.

We look forward to receiving your revised manuscript.

Kind regards,

Geofrey Musinguzi, MPH, PhD

Academic Editor

PLOS ONE

Journal Requirements:

Reviewers' comments:

Reviewer's Responses to Questions

**Comments to the Author**

1. If the authors have adequately addressed your comments raised in a previous round of review and you feel that this manuscript is now acceptable for publication, you may indicate that here to bypass the “Comments to the Author” section, enter your conflict of interest statement in the “Confidential to Editor” section, and submit your "Accept" recommendation.

Reviewer #1: All comments have been addressed

Reviewer #2: (No Response)

2. Is the manuscript technically sound, and do the data support the conclusions?

Reviewer #1: Yes

Reviewer #2: Partly

3. Has the statistical analysis been performed appropriately and rigorously? 

Reviewer #1: Yes

Reviewer #2: Yes

4. Have the authors made all data underlying the findings in their manuscript fully available?

Reviewer #1: Yes

Reviewer #2: Yes

5. Is the manuscript presented in an intelligible fashion and written in standard English?

Reviewer #1: Yes

Reviewer #2: Yes

6. Review Comments to the Author

Reviewer #1: 1. The Tables are very lengthy and provide unnecessary information.

2. I would suggest that in each row, in Tables 1 and 2, the authors only retain the Yes (risk factor, positive findings) row and delete the No row. This would make the Tables less cluttered and easier to comprehend.

Reviewer #2: Manuscript Number: PONE-D-21-20756R2

Association of lipoprotein(a) with coronary artery disease in a South Asian population: A case control study

PLOS ONE

Thank you once more. The authors addressed most of my concerns. However, I still have issues that need to be addressed. Specific comments and questions are provided below.

1.The authors’ focus shifted from a kind of prediction research to “Association of lipoprotein(a) with coronary artery disease”. Thus, the study approach, statistical analysis and interpretation of results would also change. There are descriptions in the manuscript that seem for prediction type of research. Here are some of the descriptions in the manuscript that confused risk factor identification research to prediction…

•First sentence of the discussion section still reads as “The study results suggest that Lp(a) has a modest predictive accuracy for CAD especially in subjects ≤50 years of age.”

•The first sentence of summary (page 13, line 13-14) also phrased as “In summary our study suggests that Lp(a) is an important predictor for CAD especially in the age group < 50 years.”

•The conclusion section of the abstract.

•Abstract line 13

•Analysis section, page 7 line 3 to 8. This description of analysis is still on prediction.

•Result page 10, lines 5 to 9: these descriptions are for prediction type of research.

•Even the justification of this study (last paragraph of the introduction) is prediction and risk stratification, which are the purpose of prediction research.

•Many more….

These sentences lead to an impression that the study is predictive. Better to revise it.

2.Result: page 8 line 5 to 6: “In cases, 39.3% had higher levels of Lp(a) as compared to 19% in controls and this was statistically significant (p=0.053)”. You set your significance level p<0.05 in the analysis. But here, P=0.053 is considered as significant. Revise the numerical inconsistency.

3.There are some grammar and spelling issues that need to be addressed to improve readability.

7. PLOS authors have the option to publish the peer review history of their article (what does this mean?). If published, this will include your full peer review and any attached files.

Reviewer #1: No

Reviewer #2: No

---

## [Author Response · Author response to Decision Letter 2]

16 Mar 2022

Sub: Reply to the editor’s and reviewers’ comments on the paper number PONE-D-21-20756R2

Dear Editor,

We appreciate the comments from the honourable Editor and the Reviewers. We are highly obliged to the journal for providing us the opportunity to revise the manuscript entitled “Association of lipoprotein (a) with coronary artery disease in a South Asian population: A case control study” and retain the opportunity to get our work published with PLOS ONE. We must mention that the comments have been very useful to improve the quality of our paper.

We have modified the manuscript as per the journals requirements. Our response to each and every point raised by the two reviewers is also being uploaded and incorporated in the revised version of the manuscript. We also request you, to notify us if any more corrections are needed.

Thanking you in advance for your sincere efforts and support.

Author’s response to the Editor’s comments

Comment: A rebuttal letter that responds to each point raised by the academic editor and reviewer(s). You should upload this letter as a separate file labelled 'Response to Reviewers_R3'.

Response: Rebuttal letter uploaded as a separate file 

Comment: A marked-up copy of your manuscript that highlights changes made to the original version. You should upload this as a separate file labelled 'Revised Manuscript with Track Changes_R3'.

Response: A marked up copy of the manuscript “PONE-D-21-20756R3_marked up” uploaded

Comment: An unmarked version of your revised paper without tracked changes. You should upload this as a separate file labelled 'Manuscript'

Response: An unmarked version of the manuscript” Manuscript_R3” uploaded

Yours sincerely,

Dr Geetha R Menon

 

Authors Response to the reviewers comments

Reviewer #1:

1. The Tables are very lengthy and provide unnecessary information.

Response: We appreciate the reviewer’s suggestion. The tables have been reduced after removing unnecessary information

2. I would suggest that in each row, in Tables 1 and 2, the authors only retain the Yes (risk factor, positive findings) row and delete the No row. This would make the Tables less cluttered and easier to comprehend.

Response:. We have reduced the ‘No’ categories for each variable. The information in tables 2 and 3 has now been combined into one table. 

Reviewer #2: Manuscript Number: PONE-D-21-20756R2 Association of lipoprotein (a) with coronary artery disease in a South Asian population: A case control study

PLOS ONE

Thank you once more. The authors addressed most of my concerns. However, I still have issues that need to be addressed. Specific comments and questions are provided below.

Response: We are glad that we have been able to address many of the reviewers concern.

1. The authors’ focus shifted from a kind of prediction research to “Association of lipoprotein (a) with coronary artery disease”. Thus, the study approach, statistical analysis and interpretation of results would also change. There are descriptions in the manuscript that seem for prediction type of research. Here are some of the descriptions in the manuscript that confused risk factor identification research to prediction…

•First sentence of the discussion section still reads as “The study results suggest that Lp(a) has a modest predictive accuracy for CAD especially in subjects ≤50 years of age.”

Response: We have now rewritten the first paragraph of the discussion section 

•The first sentence of summary (page 13, line 13-14) also phrased as “In summary our study suggests that Lp(a) is an important predictor for CAD especially in the age group < 50 years.”

Response: We have rewritten the first sentence of the summary section

•The conclusion section of the abstract.

Response: Response: We have rewritten the conclusion section of the abstract.

•Abstract line 13

Response: We have rewritten the abstract line 13

•Analysis section, page 7 line 3 to 8. This description of analysis is still on prediction.

Response: The analysis section has been rewritten as an association study

•Result page 10, lines 5 to 9: these descriptions are for prediction type of research.

Response: The results section has been rewritten as an association study

---

## [Decision Letter · Decision Letter 3]

18 Apr 2022

Association of lipoprotein(a) with coronary artery disease in a South Asian population: A case control study

PONE-D-21-20756R3

Dear Dr.Ravindran,

We’re pleased to inform you that your manuscript has been judged scientifically suitable for publication and will be formally accepted for publication once it meets all outstanding technical requirements.

Kind regards,

Geofrey Musinguzi, MPH, PhD

Academic Editor

PLOS ONE

Additional Editor Comments (optional):

Reviewers' comments:

Reviewer's Responses to Questions

**Comments to the Author**

1. If the authors have adequately addressed your comments raised in a previous round of review and you feel that this manuscript is now acceptable for publication, you may indicate that here to bypass the “Comments to the Author” section, enter your conflict of interest statement in the “Confidential to Editor” section, and submit your "Accept" recommendation.

Reviewer #1: All comments have been addressed

2. Is the manuscript technically sound, and do the data support the conclusions?

Reviewer #1: Yes

3. Has the statistical analysis been performed appropriately and rigorously? 

Reviewer #1: Yes

4. Have the authors made all data underlying the findings in their manuscript fully available?

Reviewer #1: Yes

5. Is the manuscript presented in an intelligible fashion and written in standard English?

Reviewer #1: Yes

6. Review Comments to the Author

Reviewer #1: No comments.

Thanks for submitting the revised version. All the comments have been incorporated in the manuscript.

7. PLOS authors have the option to publish the peer review history of their article (what does this mean?). If published, this will include your full peer review and any attached files.

Reviewer #1: No

---

## [Editor Report · Acceptance letter]

22 Apr 2022

PONE-D-21-20756R3 

Association of lipoprotein (a) with  coronary artery disease in a South Asian population: a case-control study 

Dear Dr. Menon:

I'm pleased to inform you that your manuscript has been deemed suitable for publication in PLOS ONE. Congratulations! Your manuscript is now with our production department. 

Kind regards, 

on behalf of

Dr. Geofrey Musinguzi 

Academic Editor

PLOS ONE